# sEMG-Based Continuous Hand Action Prediction by Using Key State Transition and Model Pruning

**DOI:** 10.3390/s22249949

**Published:** 2022-12-16

**Authors:** Kaikui Zheng, Shuai Liu, Jinxing Yang, Metwalli Al-Selwi, Jun Li

**Affiliations:** 1School of Advanced Manufacturing, Fuzhou University, Quanzhou 362200, China; 2Quanzhou Institute of Equipment Manufacturing, Chinese Academy of Sciences, Quanzhou 362216, China

**Keywords:** sEMG, GMM-HMMs, key state transition, model pruning, continuous two-hand action prediction, sliding window

## Abstract

Conventional classification of hand motions and continuous joint angle estimation based on sEMG have been widely studied in recent years. The classification task focuses on discrete motion recognition and shows poor real-time performance, while continuous joint angle estimation evaluates the real-time joint angles by the continuity of the limb. Few researchers have investigated continuous hand action prediction based on hand motion continuity. In our study, we propose the key state transition as a condition for continuous hand action prediction and simulate the prediction process using a sliding window with long-term memory. Firstly, the key state modeled by GMM-HMMs is set as the condition. Then, the sliding window is used to dynamically look for the key state transition. The prediction results are given while finding the key state transition. To extend continuous multigesture action prediction, we use model pruning to improve reusability. Eight subjects participated in the experiment, and the results show that the average accuracy of continuous two-hand actions is 97% with a 70 ms time delay, which is better than LSTM (94.15%, 308 ms) and GRU (93.83%, 300 ms). In supplementary experiments with continuous four-hand actions, over 85% prediction accuracy is achieved with an average time delay of 90 ms.

## 1. Introduction

The surface electromyography (sEMG) signals, as electrical signals reflecting neuromuscular function, can produce 30–150 ms earlier than the movement of hand action [1]. It contains useful information about human motion intention and is extensively used to control prosthetic hands [2,3,4]. There has been considerable interest in recognizing human motion intention by sEMG signals in recent years.

Two main aspects of applications have been implemented: discrete motion classification and continuous joint motion estimation [5]. Discrete motion classification is used to recognize specific gestures of the upper limb [6,7,8,9] and lower limb [10,11,12,13] without strict constraints on the extent of motion.

Relevant works on motion classification emphasize using appropriate features and classifiers to enhance recognition accuracy. Narayan et al. [14] used a discrete wavelet transform (DWT) to extract the time–frequency features of sEMG signals. SVM was applied to classify six gestures with the features. The results showed that 95.8% accuracy could be achieved. Liu et al. [15] used the autoregressive power spectrum to recognize 13 different gestures, and the average accuracy was up to 95%. In another research study, Yang et al. [16] proposed that RMS values could be used to build feature vectors. After the division of substates for HMM, a GMM-HMMs model was proposed to distinguish six hand motions; the classification accuracy was more than 95% in eight-channel sEMG. Deep learning methods have also been widely studied [17,18]. Shen et al. [19] proposed a method to complete hand motion classification based on a convolutional neural network and stacking ensemble learning. The experimental results on the NinaPro DB5 dataset [20] outperformed other methods. The average accuracy was 11.5%, 13.6%, and 10.1% higher than LDA, SVM, and LCNN, respectively. Hu et al. [21] used an attention-based hybrid CNN and RNN architecture to capture better temporal properties of electromyogram signals for gesture recognition problems. A new sEMG image representation method that enabled deep learning architectures to extract implicit correlations between different channels for sparse multichannel electromyogram signals was proposed. The recognition accuracies on different datasets were higher than the state-of-the-art performance.

Although discrete motion classification is widely investigated in the field of sEMG-based pattern recognition, it is still hard to satisfy the real-time requirements efficiently. One reason is that it does not consider the factors which affect realistic scenarios, such as the discontinuity of the discrete motion. Generally, human-hand action is a continuous process for practical applications, and continuous motion estimation is more applicable for hand action recognition.

Currently, continuous joint motion estimation is the primary research direction in continuous motion estimation, which realizes intention recognition by establishing a mapping relationship between sEMG signals and a continuous variable such as joint angle or joint torque. Motion estimation plays an essential role in stable control for robots [22,23,24,25]. In recent studies, two methods have mainly been used: one is based on the biomechanical model [26], and the other is concentrated on the regression model [27]. Research based on the biomechanical model has been widely considered [28,29,30]. For example, Ao et al. [31] proposed a method based on combining the Hill muscle model and a linear scale model to control the ankle exoskeleton robot. The method improved the performance of human–robot cooperation control for an ankle power-assisted exoskeleton robot. Durandau et al. [32] used person-specific neuromechanical models to estimate biological ankle joint torques. The method provided a new solution for systematically adopting wearable robots. Although the biomechanical model has been widely validated in the control of robots, it still cannot satisfy the requirements for robot joint flexibility and precise positioning in terms of recognition accuracy and real-time performance. In order to realize a real-time and precise estimation, regression models, such as neural network models, multivariate linear models, and polynomial models have been proposed [33,34,35]. Gui et al. [36] and Artemiadis et al. [22] confirmed that regression models implemented the real-time control of the limb exoskeleton.

These methods for continuous joint motion estimation only consider the continuous motion process of the upper and lower limbs. Similar to the upper and lower limb movements, hand actions are also continuous, and it is necessary to explore the continuous process mechanism of hand actions. Furthermore, the accurate prediction of continuous hand actions based on this process mechanism can recognize the subsequent gesture in advance, which is conducive to the coherent execution of continuous hand actions.

In our study, a method for sEMG-based continuous hand action prediction is proposed. The overview of the motion prediction is described in Figure 1. Our method was based on the GMM-HMMs model, which assumed the time-varying properties of the Markov chain. Six states were divided to explain the process of continuous hand actions. The key state transition was chosen as the prediction condition to realize the advanced prediction of hand actions. Sliding windows with long-term memory were used to simulate the prediction process. Eight subjects were recruited to perform twelve continuous two-hand actions and four continuous four-hand actions, and every continuous two-hand action was modeled. As a result, twelve models needed to be used in the prediction task. Considering the processing time and mutual interference of multimodel recognition, we set four different groups while each group containing three models starting with the same hand gesture. Then, model pruning was used to bind the specific group based on the degree of model-fitting. Implementing model pruning could decrease the number of models and reduce crosstalk between models. Compared with previous studies on discrete motion classification and continuous joint motion estimation, our study supplemented the research on sEMG-based continuous hand action prediction and provided a new idea for future work.

The main contributions of this paper are as follows:A new supplement to the research of sEMG-based motion intention recognition.A modified Viterbi algorithm of GMM-HMMs which can build long-term memory for the prediction process.A Model pruning which can expand the number of participating hand gestures for continuous multihand action prediction.

Section 2 describes the GMM-HMMs model, where the key state transition, sliding windows with long-term memory, and the model pruning are presented elaborately. Section 3 introduces experimental setups for twelve continuous two-hand actions and four continuous four-hand actions. Experimental results and discussions of continuous two-hand actions, continuous four-hand actions, and baseline results are presented in Section 4. Section 5 concludes the paper.

## 2. Methods

### 2.1. GMM-HMMs

Different from discrete motion classification, the prediction of continuous hand actions needs more information on time dependence. The hidden Markov model is applicable for time series modeling [37]. By assuming a Markov random process, the relationship between hidden states and observation can be linked, and the continuous connection can be further developed. In the study, we propose a model based on GMM-HMMs to analyze the continuous two-hand actions. RMS values were extracted as features for GMM-HMMs and divided into 6 parts. Different GMM-HMMs of continuous two-hand actions were built. An HMM can be defined as: (1)λ=(A,B,π)
where *A* is the transition probability matrix, *B* is the observation probability matrix, and π is the initial state distribution. The hidden states can be defined as *s* = [$s_{1}$,$s_{2}$,...,$s_{N}$] where *N* is the number of hidden states. In our work, *N* = 6, and $s_{1}$ refers to hand transition (rising), $s_{2}$ refers to hand stabilization, $s_{3}$ refers to hand transition (declining), $s_{4}$ refers to hand transition (rising), $s_{5}$ refers to hand stabilization, and $s_{6}$ refers to hand transition (declining). *A* = $\left\{a_{ij}\right\}$ defines the probability of transferring from state *i* at time *t* to state *j* at time *t* + 1. *B* = $\left\{b_{i}\left(O_{t}\right)\right\}$, 1≤i≤N defines the probability that state *i* generates $O_{t}$ at time *t*, where $O_{t}$ is a vector of feature values at time *t*. The initial state distribution, π = $\left\{\pi_{i}\right\}$, defines the probability of being in state *i* at the start time point. The states of HMMs were modeled by a Gaussian mixture model.

### 2.2. Key State Transition and the Marginalization of Sliding Windows

The results should be achieved before hand actions are completed for the ideal prediction of continuous hand actions. Typically, a robust and reliable judging condition for prediction is essential. Continuous two-hand actions were divided into six states, and the fourth state, which played a vital role in linking the front and rear gestures, could be used as a key state transition for prediction. During continuous hand actions, the period of hand transition is much shorter than that of hand stabilization, and the overall real-time performance can be improved if the correct prediction results can be obtained during the period of hand transition. Relevant research has shown that prediction results can be obtained during hand transition [38]. It indicates that the application of key state transition can predict the subsequent hand gesture in advance.

After providing the judging condition for continuous hand action prediction, RMS values were sequentially input into the trained models, and then the path of the key state transition could be backtracked by maximum state probability [39]. Many factors still constrained the prediction process, especially its processing time. Although we preprocessed the sEMG signals, the features were still too long to input into the model. Moreover, continuous hand action prediction aims to recognize the next gesture in advance, instead of recognizing the whole process. Unlike classification tasks that used complete features to recognize hand actions, our study utilized a sliding window method to update features for a dynamic prediction. Generally, the sliding window is mainly used to limit the time steps of RMS values and simulates the prediction process. After inputting RMS values in a sliding window to the models, the path of states in these steps could be backtracked by the Viterbi algorithm [37]. Since the sliding window could only record the state path information of the current window, we used the marginalization of sliding windows to establish long-term memory between the front and rear sliding windows. Moreover, the sliding window method could realize dynamic prediction through the relational information between windows. Overall, the specific implementation was based on the Viterbi algorithm. The Viterbi algorithm, which can decode the key state transition, is expressed as follows: (2)$$\delta_{t}\left(i\right)\;=\;\underset{i_{1},i_{2},\cdots i_{t-1}}{max}\;P\left(i_{t}=i,i_{1},i_{2},\cdots i_{t-1},O_{t},O_{t-1},\cdots O_{1}|\lambda \right),i=1,2,\cdots N$$
where $\delta_{t}\left(i\right)$ denotes the maximum probability of state paths with the state *i* at time *t*; λ denotes the setting of GMM-HMMs; $O_{t}$ denotes the observation at time *t*; and *N* denotes the number of states.

To derive the global state path, the state recursion was expressed as follows: (3)$$\psi_{t}\left(i\right)\;=\;\underset{1\leq j\leq N}{argmax}\;\left[\delta_{t-1}\left(j\right)a_{ji}\right],i=1,2,\cdots N$$

With the description of the Viterbi algorithm, we provide a modified Viterbi algorithm to realize the marginalization; the specific marginalization based on the sliding windows was expressed as: (4)$$\pi_{i}\;=P\left(s\left(1\right)=s_{i}\right)=\left\{\begin{array}{ccc} 1, & i=1\\ \; & \; & i=1,2,3,\cdots ,N\\ 0, & otherwise \end{array}\right.$$

The maximum probability of the state in the first sliding window was calculated as: (5)$$\delta_{t}\left(i\right)\;=\left\{\begin{array}{ccc} \pi_{i}b_{i}\left(o_{1}\right), & t=1\\ \; & \; & i=1,2,\cdots ,N\\ \underset{1\leq j\leq N}{max}\;\left[\delta_{t-1}\left(j\right)a_{ji}\right]b_{i}\left(o_{t}\right), & 2\leq t\leq T \end{array}\right.$$

The maximum probability of the state in the subsequent sliding windows was modified as: (6)$$\delta_{t}\left(i\right)\;=\left\{\begin{array}{ccc} \underset{1\leq j\leq N}{max}\;\left[\delta_{T}\left(j\right)a_{ji}\right]b_{i}\left(o_{t}\right), & t=1\\ \; & \; & i=1,2,\cdots ,N\\ \underset{1\leq j\leq N}{max}\;\left[\delta_{t-1}\left(j\right)a_{ji}\right]b_{i}\left(o_{t}\right), & 2\leq t\leq T \end{array}\right.$$

When *t* is equal to 1, the last state probability distribution of the previous sliding windows was used to initialize the subsequent sliding windows.

The following formula could backtrack the path of the state transition: (7)$$\psi_{t}\left(i\right)\;=\left\{\begin{array}{ccc} 0, & t=1\\ \; & \; & i=1,2,\cdots ,N\\ \underset{1\leq j\leq N}{argmax}\;\left[\delta_{t-1}\left(j\right)a_{ji}\right], & 2\leq t\leq T \end{array}\right.$$
where $b_{i}\left(o_{t}\right)$ denotes the observation probability density in state *i* at time *t*; $a_{ji}$ represents the transition probability from state *j* to state *i*; $\pi_{i}$ is the initial state probability distribution; and *T* is the length of the sliding window.

Generally, the marginalization is based on the inheritance of constraints on the state probability distribution of previous sliding windows. By utilizing $\delta_{t}\left(i\right)$ of the last time step in the previous window as the initialization of the subsequent window, long-term memory between sliding windows was realized. The specific illustration is shown in Figure 1.

### 2.3. Model Pruning

The number of models is proportional to that of hand gestures in GMM-HMMs-based gesture recognition [40]. For real-time prediction tasks, inputting features into multiple models is bound to be time-consuming. In addition, many irrelevant models may also interfere with the prediction results. In order to reduce the mutual interference of multiple models on recognition results and real-time performance, we propose a method for model pruning. It was noted that model pruning could be used in continuous multigesture action prediction.

After constructing a set of GMM-HMMs on continuous two-hand actions, we could find the most probable model by using the maximum likelihood criterion with the distribution of the observation sequence [16]. During the configuration of model pruning, we applied that criterion to locate pruning models. The process of model pruning could be generalized into three steps. In order to facilitate further description, we set the front gesture in continuous two-hand actions as $h_{f}$ and the other as $h_{r}$. Four different groups corresponding to four hand gestures were set, and each group contained three models starting with the same hand gesture.

Firstly, RMS values in the sliding window were input into multiple models, and then the state path with the maximum probability could be derived. It was noted that the maximum likelihood varied cumulatively using the marginalization of sliding windows. With the movement of the sliding window, the maximum likelihood would vary based on the degree of model-fitting. We searched for the key state transition in the model which had the maximum likelihood.

Secondly, the next hand action $h_{r}$ could be predicted based on the best-fitting model and the key state transition. We could bind the associated models by searching for the trained models, which were modeled by the $h_{r}$ group. Selected models could be determined from the whole model, and the number of models in the prediction task could be reduced.

Finally, the key state, which was the fourth state of the $h_{f}$ group, could be used as the first state of the $h_{r}$ group. In order to collaborate with this setting, the new starting point of the sliding window was also updated to the location of the key state. Furthermore, we could realize better predictions with fewer models. The detail of model pruning is shown in Figure 2 and Algorithm 1 is described as follows.

**Algorithm 1:** Model Pruning

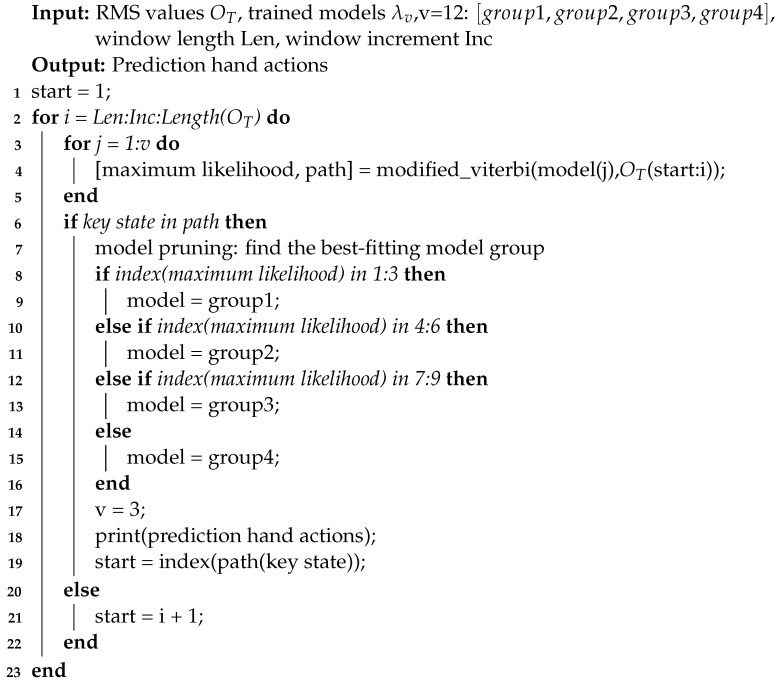



## 3. Materials and Experimental Methods

### 3.1. Experiment Setup

The datasets from previous works [20,41,42] only include the sEMG of discrete hand gestures, which are not applicable to continuous hand action prediction. Our study created a new dataset which was composed of the sEMG of continuous two-hand actions and continuous four-hand actions. The new dataset could be used for continuous hand action prediction.

Eight right-handed subjects (aged 24 to 30) participated in the experiment. Four distinct hand gestures were chosen to form twelve types of continuous two-hand actions, and each type of continuous two-hand action was formed by the continuous action of two different hand gestures. Figure 3 shows all twelve continuous two-hand actions, and four hand gestures are listed as follows: (1) making a fist, (2) fingers spread, (3) wrist extension, and (4) wrist flexion.

The GForcePro+ armband system was used to record sEMG signals. Every continuous two-hand action was performed 100 times per person. Every continuous two-hand action lasted about 5 s, of which the front and rear gestures each lasted 2.5 s. In order to reduce the negative impact on muscle fatigue, participants were given a 5 s rest between every two repetitions. Besides, participants had to relax for 5 min after completing one continuous two-hand action with twenty repetitions. The recorded sEMG signals were collected at a frequency of 1000 Hz and analyzed using MATLAB 2020b. The configuration of the dataset is shown in Table 1.

### 3.2. Data Preprocessing

Collected sEMG signals were preprocessed using a bandpass filter (20 Hz–150 Hz) to remove unusable noise and a notch filter to remove 50 Hz power disturbances. An energy-based active segmentation algorithm was used to segment sEMG signals for accurate data acquisition [16]. On the filtered sEMG signals of continuous two-hand actions, we applied overlapping sliding windows to extract RMS values. To fully use the latent information in sEMG signals and reduce the processing time of long sEMG segment length, we set a 100 ms window length with a 50 ms window increment for sliding windows.

### 3.3. Training and Prediction Setup

In the experiment, a six-state fully connected left-to-right structure HMM was used. According to the setting of continuous two-hand actions, six states were divided into two parts, while each of them contained three states: transitive state (rising), steady state, and transitive state (declining). Since all the hand actions started from state one, the initial state probability of state one was set to 1, and those of other states were uniformly set to 0. We adopted a single-component GMM to represent the substate of HMMs, as Yang et al. [16] indicated that it was less time-consuming, and it could obtain a high recognition accuracy during modeling. For 12 continuous two-hand actions, every continuous two-hand action was modeled using 100 samples per person, and a total of 1200 samples were used to construct the whole model. Every model adopted 80% samples as training data and 20% as validation data. To avoid overfitting, the training and validation data were randomly selected and modeled ten times. The final prediction result was the mean of the ten prediction results.

In order to validate the feasibility of the setting on the key state transition, the data before states 4 and 5 were split for prediction. A sliding window with a length of 20 frames and an increment of 20 frames was used. As part of continuous multi-gesture actions, continuous four-hand actions were selected as being representative for prediction. Since continuous four-hand actions had plenty of combinations, we chose four different types for the experiment. For continuous four-hand actions, we still used RMS values as the input of the models. It was noted that the acquisition time needed to be limited to 10 s. We collected 30 samples for each type of continuous four-hand actions to predict.

The proposed method was used to predict continuous hand actions in advance; the special partitioning and modeling of continuous states made it better than other methods. To show the robustness of our method, we used conventional LSTM and GRU models, which have been widely utilized in discrete gesture prediction for comparison. In order to ensure the uniformity of the comparison, we also used the sliding window strategy on LSTM and GRU to simulate the prediction process. A sliding window with a length of 20 frames and an increment of 20 frames was used.

## 4. Results and Discussion

### 4.1. Estimation of Continuous Two-Hand Actions

#### 4.1.1. Validation on the Setting of Key State Transition

Figure 4 demonstrates the prediction accuracy. We found that the prediction accuracy of state four and state five were all higher than 95%, which indicated that utilizing state four and state five as the key state transitions could predict hand actions correctly. Although the prediction accuracy of state five as the key state transition was higher than state four, the occurrence of state four was earlier than that of state five. It was more efficient to use state four as the key state transition to make the advanced prediction.

#### 4.1.2. Validation on the Marginalization of Sliding Windows

We investigated the effect of different sliding window settings on the prediction results. Figure 5 illustrates the prediction accuracy of four different sliding window settings. It could be observed that all four different settings achieved over 92% recognition accuracy, which indicated that different settings of the sliding window had little effect on the results and proved that the marginalization of sliding windows could help us build long-term memory between sliding windows, which guaranteed the prediction accuracy. Furthermore, it allowed us to choose the appropriate sliding window setting based on demand instead of a fixed setting. We also analyzed the reasons for the different prediction results of four settings. In our study, we gave the prediction results as soon as the key state transition was found. Influenced by the size of the sliding window, the predicted time point of finding the key state transition was different, resulting in the difference between the four settings. We also assumed that selecting a reasonable size of the sliding window based on the length of RMS values would reduce the time-consuming for prediction.

#### 4.1.3. Comparison with Other Methods

The normalized confusion matrix of our method is shown in Figure 6. In order to lift the limitation of LSTM and GRU [38] on the input dimension, we fixed the input starting point and used a sliding window to expand the data successively, in which the insufficient data were filled with zeros. The results are depicted in Figure 7.

Table 2 shows the average prediction accuracy and the processing time for all subjects. As seen in Table 2, our proposed method achieved a better prediction accuracy and was less time-consuming than other methods. LSTM and GRU had a good prediction performance but unacceptable processing time. Although the same sliding-window strategy was used in LSTM and GRU, the real-time performance of these methods was poor. The main reason was that no matter whether the sliding window was updated, the input dimension was always limited to the maximum. Compared with these methods, our method only utilized the updated data from the sliding window as input for the prediction, which guaranteed less processing time.

### 4.2. Estimation of Continuous Four-Hand Actions

Continuous four-hand action prediction was tested to evaluate the performance of model pruning. The prediction results could also validate the feasibility of continuous multigesture action prediction. Since there were many combinations of continuous four-hand actions, we selected four types of them which are listed as follows: (1): making a fist–fingers spread–wrist flexion–making a fist; (2): wrist flexion–wrist extension–making a fist–fingers spread; (3): fingers spread–making a fist–wrist flexion–wrist extension; (4): wrist extension–fingers spread–making a fist–wrist flexion.

Table 3 shows the comparison between model pruning and no model pruning. The average prediction accuracy when using model pruning was over 92.9%, while that without model pruning was only over 20%. The processing time with no model pruning was too poor to satisfy the real-time performance requirement, while model pruning could tremendously reduce the processing time. The prediction results of subjects are also illustrated in Table 4. These results indicated that our proposed model pruning method achieved a significant prediction accuracy and better real-time performance. Since our trained models were modeled by continuous two-hand actions and depended on the accumulation of relevant state probabilities, they could only be used to predict continuous two-hand actions. Surprisingly, when the RMS values exceeded the period, the accumulation of state probabilities was irrelevant to the subsequent hand gesture and would affect the subsequent prediction process, which resulted in a terrible performance without model pruning. These findings suggested that model pruning was necessary and should be utilized in the range of the period. We updated the starting point of state probability accumulation and selected the best-fitting models to predict the subsequent gesture, which could also reduce the processing time of the model prediction.

The prediction accuracy of continuous four-hand actions indicated that model pruning could apply to our work. Notably, as a part of continuous multigesture action prediction, the excellent prediction accuracy of continuous four-hand actions further confirmed the feasibility of continuous multigesture action prediction with our method.

### 4.3. Limitations

Currently, our study focuses on predicting the combinations of four distinctive hand gestures. The prediction task of the combinations of fine-hand actions has not been carried out yet. Moreover, subject-independent problems still exist in our method, and every subject must be modeled separately in the experiment. Our future work will concentrate on predicting continuous fine-hand actions and providing a robust universal model which can solve subject-independent problems.

## 5. Conclusions

In this paper, we proposed a continuous hand action prediction method based on the continuity of hand actions. In order to predict the subsequent hand gesture in advance, our method set the key state transition as the prediction condition. The GMM-HMMs model was utilized to model the states of continuous two-hand actions, and state four was set as the key state transition. The marginalization of sliding windows was used to build long-term memory for windows, and a dynamic sliding window was used to simulate the prediction process. Compared with LSTM and GRU models, our method achieved a higher prediction accuracy and a lower processing time for continuous two-hand actions. In the extended task of continuous multigesture action prediction, the GMM-HMMs models built by continuous two-hand actions could not predict continuous hand actions with more than two hand gestures. We proposed a model-pruning method which selected the best-fitting models and updated the starting point of state probability accumulation to overcome the limitation. The prediction results also confirmed the feasibility of the method on continuous multigesture action prediction.

## Figures and Tables

**Figure 1 sensors-22-09949-f001:**
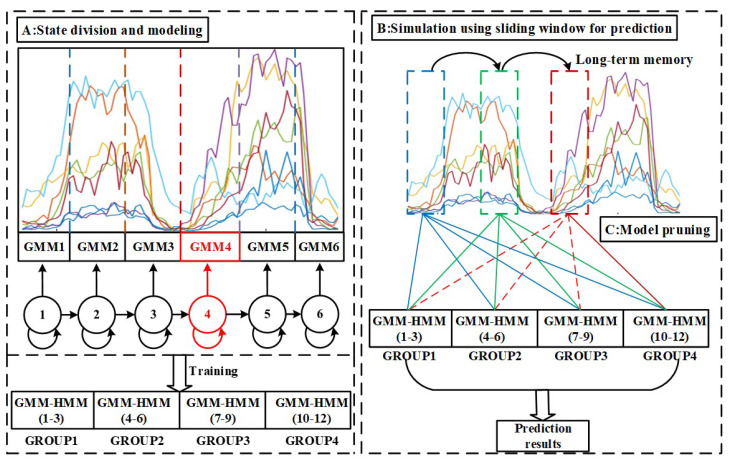
The structure of the proposed prediction framework. Part A illustrates the state division and modeling process; state 4 is the key state transition of the setting. Twelve models are trained and then divided into four groups according to the different starting gestures of continuous two-hand actions. Part B displays the prediction simulation process using a sliding window with long-term memory. Based on Part A and Part B, Part C performs model pruning; the groups pointed by the dotted line are pruned.

**Figure 2 sensors-22-09949-f002:**
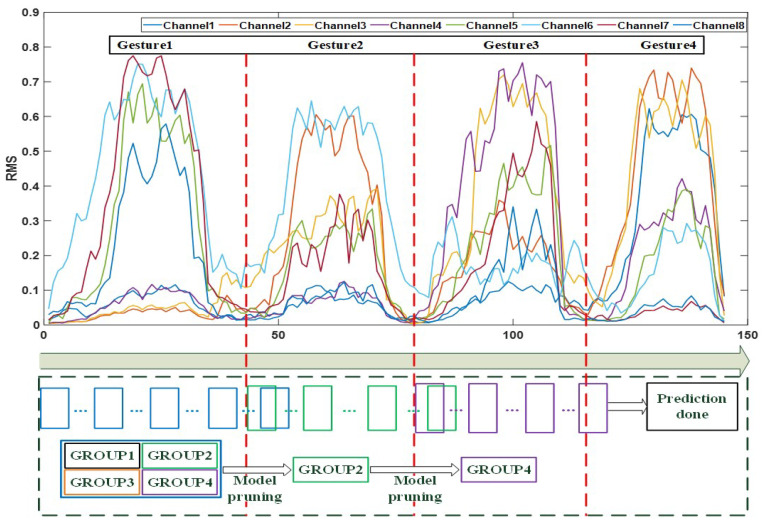
Model pruning with sliding windows on continuous four-hand actions. Every box represents a dynamic sliding window without overlapping. The whole model was divided into 4 groups, while blue sliding windows used the whole model for prediction. Green and purple sliding windows used GROUP2 and GROUP4 for prediction, respectively. The red dotted line represents the starting point of the new hand gesture, which can be used for updating the new starting point of the sliding window in model pruning.

**Figure 3 sensors-22-09949-f003:**
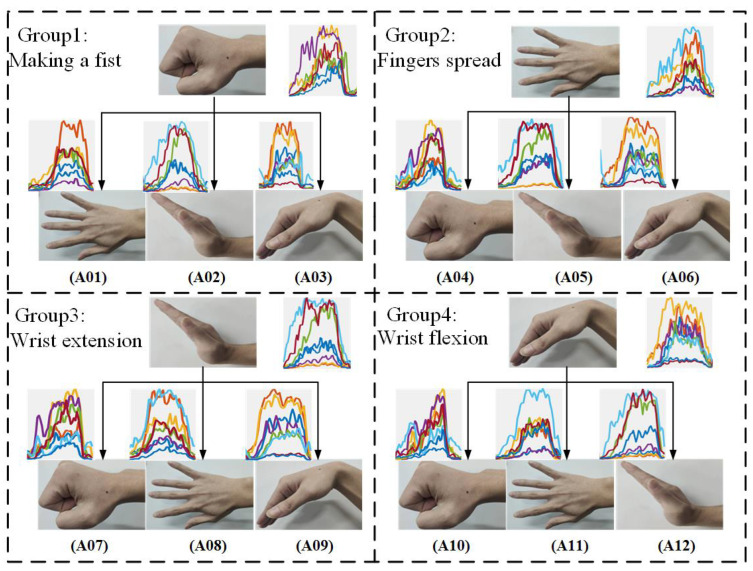
Illustration of continuous two-hand actions; four groups are determined by different starting hand gestures.

**Figure 4 sensors-22-09949-f004:**
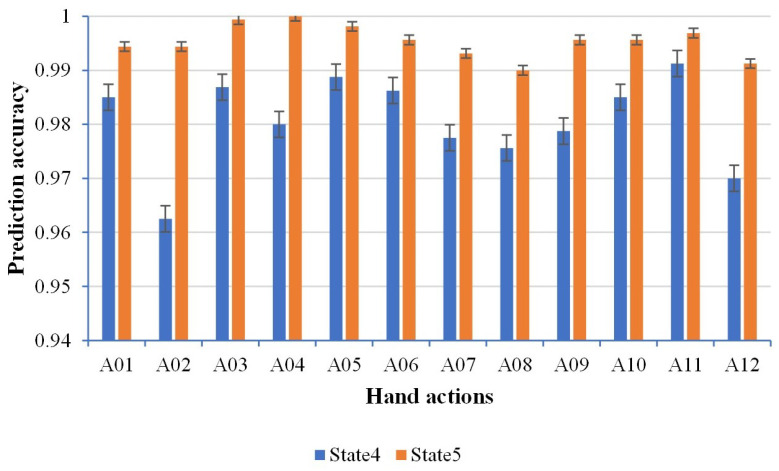
Prediction accuracy for two different settings of key state transition.

**Figure 5 sensors-22-09949-f005:**
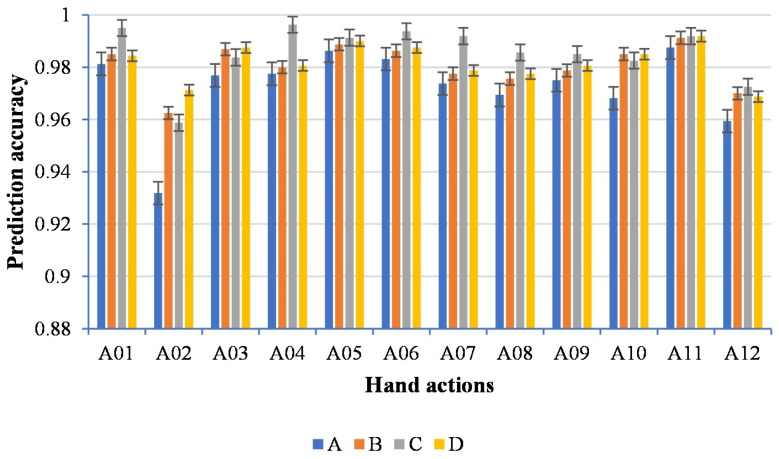
Prediction accuracy for four different settings of the sliding window. A represents a length of 10 frames with an increment of 10 frames; B represents a length of 20 frames with an increment of 20 frames; C represents a length of 25 frames with an increment of 25 frames; D represents a length of 30 frames with an increment of 30 frames.

**Figure 6 sensors-22-09949-f006:**
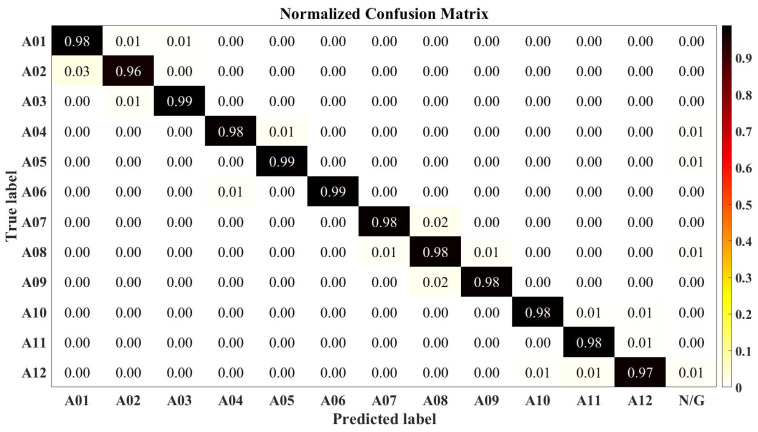
Hybrid test confusion matrix of all subjects with our method. N/G represents nongesture.

**Figure 7 sensors-22-09949-f007:**
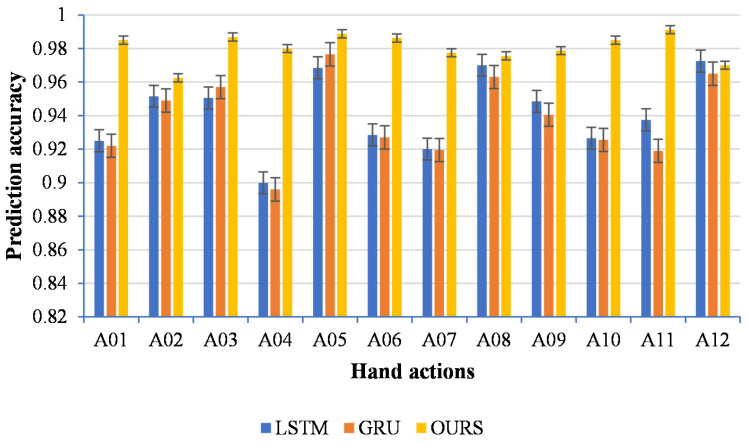
Prediction accuracy compared with LSTM and GRU.

**Table 1 sensors-22-09949-t001:** Configuration of the dataset.

Acquisition Device	GForcePro+	Sampling Frequency	1000 Hz
Number of channels	8	Number of subjects	8
Age range of subjects	24–30	Health state	Intact subjects
Type	Continuous 2	Continuous 4	
	hand actions	hand actions	
Hand actions	12	4	
Repetition times	20	10	
Sampling time of a repetition	5 s	10 s	
Repetition interval	5 s	10 s	
Number of repetitions	5	3	
Action interval	5 min	5 min	

**Table 2 sensors-22-09949-t002:** Average prediction accuracy and processing time of different methods for all subjects.

Metric		Subjects	S1	S2	S3	S4	S5	S6	S7	S8
	Method									
Prediction accuracy (%)	LSTM		97.9	95.8	94	91.6	95.0	93.1	93.9	91.9
	GRU		97.3	97.8	94.3	92.1	94.9	90.7	93.9	89.6
	OURS		99.7	98.6	98.9	96.4	99.3	95.4	99.5	96.6
Processing time (ms)	LSTM		303	300	300	323	310	307	315	313
	GRU		293	290	297	320	301	295	305	297
	OURS		71	69	68	72	69	73	71	72

**Table 3 sensors-22-09949-t003:** Comparison between model pruning and no model pruning.

Metric		Hand Actions	(1)	(2)	(3)	(4)
	Method					
Prediction accuracy (%)	Model pruning		95.8	92.9	94.6	95
	No model pruning		31.7	22.5	20	21.3
Processing time (ms)	Model pruning		96	95	97	94
	No model pruning		272	281	269	277

**Table 4 sensors-22-09949-t004:** Average prediction accuracy and processing time of continuous four-hand actions.

Metric		Subjects	S1	S2	S3	S4	S5	S6	S7	S8
	Hand Actions									
Prediction accuracy (%)	(1)		96.7	96.7	96.7	83.3	96.7	100	100	96.7
	(2)		100	96.7	93.3	86.7	100	90	86.7	90
	(3)		100	86.7	96.7	90	100	96.7	93.3	93.3
	(4)		100	96.7	90	80	100	100	100	93.3
Processing time (ms)	(1)		94	89	93	92	91	95	93	95
	(2)		94	92	94	92	90	94	93	94
	(3)		96	94	94	95	93	94	92	95
	(4)		93	91	97	94	92	97	94	96

## Data Availability

Data sharing is not applicable to this article. Please contact the authors for further requests.

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
