# Peer review of "sEMG-Based Continuous Hand Action Prediction by Using Key State Transition and Model Pruning"

_sensors, 2022, doi:10.3390/s22249949_

Round 1

Reviewer 1 Report

For the issue that few researchers have investigated continuous hand action prediction based on hand motion continuity, in this paper, authors proposed key state transition as a condition for continuous hand action prediction and simulate the prediction process using a sliding window with long-term memory. Their method achieved relatively high prediction accuracy and low time-consuming for continuous two-hand actions.However, to be a good paper, there are some major modifications have to be made carefully.

1          Since model pruning has been done, is there any experiment comparison before and after pruning?

2          Table 1: a mixture of three and four columns, which is easy to cause misunderstanding without looking carefully.

3          Some sentences in the article need to be carefully pondered and polished. For example, on line 315, page 12: “In this paper, we proposed continuous hand action prediction based on the continuity of hand actions.” The proposal is suitable for a method, not a prediction.

Reviewer 2 Report

The manuscript presents a method for hand action prediction using surface electromyography signals.

The paper is well-written, and the methodology and results look good. However, the paper lacks novelty. The methods used GMM-HMM are old and well-known, which should be a baseline approach, not the final one.

The dataset where the method was trained and tested is very small to have any statistical significance on the results. Other works [1, 2] collect larger datasets for the same purpose. Why collect a new one that is no better?

The comparisons with other methods are not enough, the authors compare with LSTM and GRU, which should also be baseline models.

I suggest comparing their method against other datasets and published state-of-the-art methods before considering publication in a Journal.

[1] Kaczmarek P, MaÅ„kowski T, TomczyÅ„ski J. putEMG-A Surface Electromyography Hand Gesture Recognition Dataset. Sensors (Basel). 2019 Aug 14;19(16):3548. doi: 10.3390/s19163548. PMID: 31416251; PMCID: PMC6720505.

[2] Ozdemir, Mehmet Akif, et al. "Dataset for multi-channel surface electromyography (sEMG) signals of hand gestures." Data in brief 41 (2022): 107921. 

Round 2

Reviewer 2 Report

Taking into account the response of the authors. I suggest two changes:

1. Add a contributions paragraph at the end of the introduction stating the innovations that they proposed and that were mentioned in the response to my comments:

Overall, the innovations of the paper are as follows: 1. A new supplement to the research of sEMGbased motion intention recognition; 2. Modified Viterbi algorithm of GMM-HMMs which can build long-term memory for the prediction process; 3. Model pruning which can expand the number of the participating hand gestures for continuous multi-hand action prediction.

2. In the database section, mention the limitations of the existing datasets and how this new dataset fill the gap.
